# The Nursing Role in the Management of Medication Overuse Headache: Realities and Prospects

**DOI:** 10.3390/brainsci14060600

**Published:** 2024-06-14

**Authors:** Luigi Alberto Pini, Katiuscia Cottafavi, Paola Ferri

**Affiliations:** Department of Biomedical, Metabolic and Neuroscience, University of Modena and Reggio Emilia, 41125 Modena, Italy; katiuscia.cottafavi@unimore.it (K.C.); paola.ferri@unimore.it (P.F.)

**Keywords:** medication overuse headache (MOH), nursing, nurse, headache

## Abstract

This review aims to analyze the current literature to identify articles related to the role of nurses and, in general, the nursing management of patients suffering from medication overuse headache (MOH), a globally spread disease. We specifically argue for non-pharmacological approaches to pain management, such as multidisciplinary team approaches, holistic treatment, cognitive behavioral therapy and exercise. For this review, we investigated international scientific databases, including PubMed, CINAHL, Scopus and Embase, in the period between 2000 and 2024. We observed a wealth of scientific articles related to MOH, but a poverty of articles relating to the nursing management of headache. The research included the presence of academic-level training for nurses, whereas there are few institutions that train competent professionals in both pharmacological and non-pharmacological management of MOH patients. Nursing assessment and assistance strategies are indicated to plan tailored treatment paths related to the specific needs of these patients.

## 1. Introduction

Medication overuse headache (MOH) is a disorder well described in the third edition of the International Classification of Headache Disorders (IHCD-3) [1]. MOH is described as a secondary headache occurring for 15 or more days per month in patients with a pre-existing headache and with regular medicinal use of painkillers for at least three months. This disorder affects 1–2% of the population, so the total prevalence worldwide is estimated to be around 63 million people, and it represents the most frequent diagnosis of hospitalized patients in headache centers [2].

MOH is generally considered a secondary headache disorder, despite pathophysiological and clinical evidence that could suggest classifying it as a primary disorder, rather as a complication or transformation of pre-existing primary headaches. On the other hand, the clinical diagnosis is well defined, and is linked to a series of symptoms that can be documented by the patients’ diaries, a method that tends to define both the frequency and intensity of the symptoms in a categorical way, which then makes it possible to compare data obtained in different countries. The distinctive parameter is the number of medications consumed monthly, specifically the regular consumption and the types of drugs that fill the diagnostic criteria for MOH. In fact, ICHD-3 beta criteria state the number and type of drugs to be consumed monthly, but it might seem that the pharmacodynamic relationship between the drugs taken for headache and the appearance of MOH could be considered to be more linked to a behavior rather than being based on a direct relationship linked to the pharmacological action of anti-migraine drugs [1].

One of the major criticisms concerns the term ‘overuse’, which is usually not translated into different languages. In fact, the term is clinically described and detailed, and one of the items necessary for diagnosis is that the consumption of analgesics must be classified as ‘regular consumption’; this represents a criterion linked to how and when analgesic drugs are consumed by patients suffering from frequent and severe headache. This diagnosis presents some differences when compared to addiction behaviors. This last term should only be used to refer to ‘addictive’ behaviors to avoid stigmatizing headache patients. It is necessary to underline the specificity of MOH and the difficulties in defining the pathogenetic model of the disease. On the other hand, the lack of well-defined biochemical targets justifies the enormous variety of treatments that have been proposed for MOH treatment, ranging from a simple warning to out-patients to hospitalization; from this evidence emerges the need to correctly manage the organization of the various therapies and, therefore, the patient’s education on dealing with complicated therapeutic paths, as summarized in the Japanese Guidelines for Chronic Headaches (CH) (https://ihs-headache.org/wp-content/uploads/2020/06/2528_japanese-headache-society-clinical-practice-guideline-2013.pdf, accessed on 28 March 2024). These Guidelines summarize in a schedule, with B-grade evidence, the treatment principles for MOH, as follows: (1) withdraw from the overused medication, (2) continue to treat headache attacks and (3) start a new prophylactic treatment. To date, unfortunately there are no unanimously recognized methods of treatment. Discontinuation of the overused medication can be carried out in outpatients, but hospitalization may be necessary for withdrawal treatment in case of severe use or of opiate-/barbiturate-containing compounds overuse. The relapse rate within five years is approximately 30%. After discontinuation, patients need further support and suitable counseling actions, and filling out a headache diary is an essential element for measuring the consumption of drugs implicated in MOH pathogenesis [3].

The care pathways for headache are multidisciplinary and should include access to a number of health services and roles. Primary patient management is based on drugs, whereas numerous other approaches are also possible, ranging from dietary to physical approaches and from behavioral to neurostimulation therapies. Many of these multidisciplinary interventions reported the key role of the nurse in managing all these activities, mainly at home. The rapid evolution of healthcare organizations in headache management by general practitioners (GPs), pharmacists and nurses, including telemedicine, requires continuous updating to optimize the use of resources which could be reduced in the near future.

Therefore, the objective of this review is to re-evaluate the recent literature on the role of nursing in the management of chronic headache by grouping the data according to the types of intervention.

## 2. Materials and Methods

The electronic databases, including MEDLINE (PubMed), CINAHL-EBSCO, SCOPUS and EMBASE, were searched for the period from 2000 to January 2024, then refreshed between 4 March and 1 May 2024. The search was conducted using the following non- Medical Subject Headings (MESH) terms: ‘Medication overuse headache’ AND ‘nursing’ OR ‘Nurse’. For each result, the full text was analyzed in order to decide on its inclusion in this review. We found 68 articles for PubMed, 8 articles selected for Scopus, 6 for EMBASE and 4 for CINAHL-EBSCO. Only research products written in English and with full text available were considered. At the end of the bibliographic research, two authors made the selection, and any doubts were resolved by consulting a third researcher. The main data of the included studies according to the research questions were extracted into an Excel^®^ file, analyzed and discussed. Finally, a narrative presentation was prepared by authors, and results were divided into the following macro-areas concerning, respectively, (1) statements to date, (2) organization in headache centers, (3) nurses, (4) pharmacological therapies, (5) physical therapies and exercise, (6) counseling and multidisciplinary treatment and (7) onabotulinum toxin A.

## 3. Results

### 3.1. Statements to Date

For decades, withdrawal from analgesic medication has been represented as the first and fundamental objective step for MOH management. There are many recommended strategies and one of the most debated problems—apart from the choice of drug to use—is when to start therapy, either before or after withdrawing from analgesic medication overuse [4].

In Danish guidelines issued in 2012, an educational program about MOH withdrawal from overused drugs followed by preventive medication was recommended [5]. Later, data obtained from a larger prospective study demonstrated that preventive drug therapy starting at the time of withdrawal and the previous treatment, which began two months after drug discontinuation, were equally effective [6,7]. A comparison of the treatment strategies showed that withdrawal was able to revert the chronic form of MOH to an episodic form in approximately two-thirds of patients [8,9]; however, in other headache centers, preventive therapy was started both simultaneously with acute withdrawal and following withdrawal from the overused medication [5,10].

In clinical practice, the relapse into chronicity and the frequent use of analgesics is recurrent, and Ljubisavljevic and colleagues took into consideration the predictors of immediate and long-term discontinuation, including within the predictors both successful discontinuation from the beginning and the presence of a low headache index within twelve months [11,12]. Patient counseling plays a significant positive role. In fact, the relapse rate of MOH in patients treated with botulinum toxin A (BT-A) who received counseling at the beginning of detoxification was about 4% and 16% after 6 and 16 months, respectively [13]. Moreover, patients who received counseling 6 months after discontinuation did not demonstrate relapse during follow-up [14]. Other studies reported that the relapse rate in unsuccessfully treated MOH after combined drugs plus psychotherapy seemed to be superior to pharmacotherapy alone [15]. Recently, calcitonin gene-related peptide (CGRP) monoclonal antibodies (CGRP mAbs) have been proven to be effective in patients with CH and MOH, in the European Headache Federation guidelines [16]. Because it is likely that the benefit–risk profiles of newer treatments will continue to develop, as can be seen from the analysis of real-world studies, the “American Headache Society”, recognizing the continuous and rapid evolution of suggested therapies, has decided to review the statement of this pathology on a regular basis and update it if the need arises, in the interest of patient care [17].

In conclusion, despite the plethora of treatments used for the management of MOH, it seems that the conclusions drawn on the occasion of the European Academy of Neurology, 2020, are still valid, as listed below [18]:➢Patient education is a fundamental step in the correct management of MOH.➢Uncomplicated patients with MOH can be managed out of hospital or clinics by general practitioners.➢Complex MOH should be managed by multidisciplinary teams that include neurologists, headache or pain management specialists, physiotherapists and behavioral psychologists.➢Patients with poor educational and socioeconomical conditions must be treated with individualized programs based on their needs and tailored to their ability to collaborate in therapeutic interventions. Particular attention should be paid to the type of drugs overused, and only drugs with proven efficacy should be used in these patients.➢Patients with MOH who do not respond to preventive therapy should undergo drug withdrawal. When overused drugs are simple analgesics, ergots or triptan, a suspension may be abrupt, total or partial. For the long-lasting use of opioids, barbiturates or tranquilizers, a slow tapering-off from these drugs is highly recommended.

In the literature, the suspension of overuse is reported as being able to be carried out in all situations, both intra- and extra-hospital. Despite these enormous methodological differences, the short-term results are similar. An accurate medical history allows us to identify the most correct approach for each patient. Even a recent National Institutes of Health (NIH) publication confirms these suggestions and identifies some lines of development in the near future for this pathology [19].

### 3.2. Professional Figures Employed in Some Tertiary Headache Centers

In a classic survey, Bhola and Goadsby analyzed the organizational characteristics in some tertiary headache centers around the world, namely in the UK, the USA, Denmark, Thailand, Australia and Brazil [20]. The size and organization of the centers were analyzed, and the prominent features were described, underlining that the optimization of the team’s work was based on constant educational processes that aimed to improve the team’s knowledge, with regular meetings in which the circulation of information was facilitated, engaging neurologists trained in headache, specialist nurses, psychologists and physiotherapists (which some centers have). Therefore, the authors concluded that tertiary headache centers demonstrate similar organizational schemes, regardless of socio-economic and cultural characteristics, in all surveyed areas [20]. See Table 1.

### 3.3. Nurses

When we analyzed the literature using the term ‘nurse/nursing’ and ‘headache’, we found only few references, scattered in lists without precise role attribution or specified activities. In this way, it is difficult to define the characteristics that should be activated for a better function of the nurse in this setting. In the past, the role of the nurse or expert trained in headache was focused mainly on the extra-hospital setting, on the evaluation of the cost–benefit ratio [29], on patient education [30], on the administration of questionnaires [31,32] or on organizing activity supervision strategies [33].

Thus, we can summarize that nurses play many roles ranging from patient education to lifestyle improvement, medication management, telephone queries, consultations and educational input in preparing patients for nerve block procedures.

The most important centers use nurses to better define their needs, with the aim of making patients understand their therapeutic plan, the need for continuous evaluation of clinical evolution to optimize resources and the definition of an effective discharge plan. In general, nurses’ activity is aimed at improving both the patient experience and the organization of clinical work, and they must be involved in developing emerging therapies and in refresher educational plans [34].

Two large pain centers with nursing teams were located at Michigan Head Pain Center and in Copenhagen, where trained nurses were trained to take complete medical histories of medications used by patients. In other centers, such as in Bangkok and Porto Alegre, these activities were carried out by doctors-in-training. In Sydney, the role of nurses was more limited, and these activities were carried out by neurologists, while the role of nursing was oriented more towards the organization of services for patients and the management of therapeutic protocols and their execution, always under the supervision of a neurology educator, who controlled learning and performance. Ultimately, the presence of a specialized nurse in the team greatly improved the organization, both in public and private care settings [20].

An evaluation of the organization of headache centers was carried out during The European Headache and Migraine Trust International Congress (EHMTIC) held in Nice in 2010; Gaul and colleagues provided a detailed report that also included the role of nursing, specifying the durations and methods of training [28]. The number of nurses has seemingly increased with the passage of time and the increase in headache centers, and it is important that this factor is documented and their contribution recognized. Examples of headache specialist nurses are also present in Europe and the establishment of the International Forum of Headache Nurses should be a stimulus to facilitate the inclusion of these professional figures in European countries.

The definition of specialized nurses varies among countries. A nurse can be defined as a person who has completed a program of basic, generalized nursing education and has been authorized by the appropriate regulatory authority to practice nursing in their country. The basic program aims to provide broad and solid training on behavior and lifestyle in nursing, for good nursing practice, for leadership and collaboration roles in work teams and for post-basic education programs. A nurse is trained and prepared to carry out their specific tasks including health education, prevention, care for physical and mental well-being and the management of disabled people of all ages in all possible locations from the family home to hospitals. Furthermore, the nurses will be trained to teach, participate in multidisciplinary healthcare team meetings, supervise and educate auxiliary staff, and, finally, be included in research programs [35].

The International Council of Nurses (ICN) defines a nurse practitioner/advanced practice nurse as a nurse who has acquired the basic specialist knowledge, complex decision-making skills and competencies for in-depth clinical practice, the characteristics of which are shaped by context and/or the country in which they practice [36]. Advanced practice is a level of practice, rather than a type of practice. Advanced nurse practitioners have master’s level training in clinical practice and have been assessed as competent in practice at using their specialist clinical knowledge and skills. They have the freedom and authority to act, making autonomous decisions in the assessment, diagnosis and treatment of patients (https://www.rcn.org.uk/Professional-Development/publications/pub-007038, accessed on 3 May 2024).

In Italy, specialized nurses must, after a three-year degree in nursing (responsible for general care), attend a first-level university master’s degree.

The nursing profession in Italy is regulated in detail by ministerial directives, since the largest share of healthcare activity is linked to the National Health Service. This type of organization guarantees an adequate professional standard but limits the possibility of implementing the roles that can be formalized in clinical procedures, mainly to the structures connected to the university. For this reason, tertiary centers in Italy follow the directives already highlighted by the Emilia-Romagna region quite precisely, but the limitation of budgets has not allowed for the diffusion of experimental procedures over time. For this reason, although there are nurses trained in headache in tertiary centers, there is no formal recognition of the title, so we do not believe that this type of organization can be generalized to other National Health Services. The implementation of these experiences is desirable if and when official international professional bodies are activated. An interesting example of nurse inclusion in research protocols comes from the University of Pavia (IT), where, in the operating procedures for controlled clinical trials, nurse roles and methods of intervention are well defined [27]. During recent years, with the increase in articles published by nurses and their participation in international congresses and conferences on headache, the fundamental role of the nurse in directly caring for the patient has been highlighted. Nurses can begin to use evidence-based strategies and advice in the field of headache to optimize their relationship with these patients who are suffering greatly and live in a situation of psychological stigma, due to their continuous use of drugs, both from their own perspective and often also due to prejudice from relatives and work colleagues. It therefore appears necessary to make the nurse’s intervention more coherent in the team that treats these patients.

Within this perspective, Rasmussen and colleagues carried out a study aimed at producing European-level indications for the management of migraine patients by nursing staff. The study applied an e-Delphi method to systematically obtain a set of shared recommendations among specialists from numerous European countries [37]. The author conclusions suggested that future studies oriented on nursing in headache can ameliorate the recommended tasks to improve the management of headache patients [38].

In an another paper, Carlsen and colleagues reported that the withdrawal notice was given by trained headache nurses, who were defined as nurses specialized in headache, cited in the research team’s organizational chart [7]. Ljubisavljevic and colleagues suggested that it is necessary to check the patients’ vital parameters as well as intervene to maintain control of the symptoms that appear during this phase. Therefore, the availability of a well-organized and trained multi-professional team, including a psychologist, is recommended to obtain a positive result in the initial phase of abstinence from analgesic drugs [11].

The primary role of the nurse specialist in headache and MOH is education and should consider a number of objectives, as follows: achieving withdrawal from overused medication, recovery from MOH and review and reassessment of headache disorder [39]. Gillies and colleagues also highlighted the educational role of the nurse in preventing the excessive use of medications in headache, reported that patients appreciated having someone listen to their concerns and concluded that nursing support can improve compliance with the treatment regimen [40]. In a prospective evaluation, Clarke and colleagues used a nurse trained in the management of headache patients in the neurology unit, capable of diagnosing migraine, tension-type headache and MOH, as well as advising general practitioners on treatment and collaborating with consultant neurologists. The nursing service eliminated waiting times longer than 13 weeks and reduced waiting times for all neurological patients [41]. A retrospective controlled study highlighted the importance of the headache-specialized nurse leading to greater compliance in the detoxification of patients suffering from headache; therefore, it has an unequivocal effect on the success of the therapy [42]. D’Antona and Matharu, in 2019, reviewed the importance of educating patients with refractory headache about lifestyle and found that biobehavioral therapy and relaxation biofeedback were effective, in both the acute and preventive treatment of headache. A global, multimodal and multidisciplinary approach was necessary [43].

In 2019, a study by Vacca and colleagues described the nurse’s role in taking the patient’s health history and performing an objective physical examination. The assessment should include gathering information on the onset, location, duration and characteristics of the migraine, as well as associated, aggravating and relieving factors. Additionally, all treatments and subsequent patient responses should be documented in the electronic medical record. Physical evaluation should include a focused neurological evaluation to identify abnormalities. Nurses should also perform medication reconciliation, including any over-the-counter (OTC) medications, to ensure patient safety and prevent possible drug interactions with prescribed medications. This article also mentioned the nurse educational role, in which providers should teach patients to identify and recognize potential migraine triggers, including emotional factors, poor sleep hygiene, environmental factors, hormonal fluctuations, use and abuse of drugs, dietary factors and alcohol, especially red wine. Nurses should encourage patients to keep a migraine diary and offer lifestyle strategies to reduce, limit or avoid migraine triggers [44].

### 3.4. Pharmacological Therapies

In recent years, numerous studies have been reported as using many drugs for the treatment of MOH, and the variety of active ingredients used is truly wide. To date, therapy with beta-blockers, calcium channel blockers, tricyclic antidepressants and anticonvulsants has been utilized. Topiramate has demonstrated particular efficacy in patients with chronic migraine. In the detoxification phase, various neuroleptics have been used with varying effectiveness [19,45].

In the last few years, CGRP mAbs have been proven to be effective in patients with chronic migraine and MOH, as set out in a statement of European Headache Federation [11]. Moreover, these drugs seem to also ameliorate cognitive performances in MOH patients [46].

Nevertheless, in these protocols, the overall nursing role, when mentioned, was restricted to the distribution of medicines, collection of side effects and completion of questionnaires.

### 3.5. Physical Therapies and Exercise

Even though experimental studies reported exercise as a trigger factor for migraine attacks, regular exercise may have a prophylactic effect on migraine frequency. This seems linked to various factors: biological factors, the biochemical pathways involved, the frequency of headaches and the type of physical stimuli applied.

Guidelines for the treatment of headaches through chiropractic treatment were explored by Bryans and colleagues, with the aim of providing evidence-based practical recommendations [47]. This systematic research found that massage was recommended for the management of patients with episodic or chronic migraine, whereas, for tension-type headache, spinal manipulation was not recommended. With regards to tension-type headache, a systematic review provided modest efficacy results even in the face of a great variety of techniques adopted [48]. Even with regard to the treatment of cervicogenic headache, the results were potentially modest and obtained with weak evidence [44]. Multimodal multidisciplinary care, for example, exercise, relaxation, stress, nutrition counseling and finally massage, were recommended for episodic or chronic headache. There was insufficient clinical data to recommend or not the use of physical exercise alone or combined for the management of these patients [47].

More recently, other authors seem to have reached different conclusions by examining the efficacy of aerobic exercises in the treatment of migraine, including chronic migraine. These exercises in randomized controlled trials seemingly led to an improvement in migraine symptoms, whereas data on the effects of flexibility, coordination and relaxation on migraine are currently insufficient to make any recommendations. Therefore, the authors concluded by prudently suggesting only a “potentially beneficial strategy” for this treatment in migraine patients [49].

A recent meta-analysis on the effectiveness of aerobic exercises towards relaxation in headache concluded that ‘Strength training exercise regimens demonstrated the highest efficacy in reducing migraine burden, followed by high-intensity aerobic exercise’ [50], but the conclusions of this review have been strongly criticized for methodological problems [51].

It will therefore be necessary, in the future, to design studies capable of providing methodologically convincing results in order to develop clinical recommendations. We can therefore conclude that physical activity does not currently seem to have the evidence necessary to reach a consensus that can be used as a recommendation, especially for chronic headache [52].

### 3.6. Counseling and Multidisciplinary Treatment

Gaul, Van Doorn and colleagues prospectively studied a series of patients treated with a multidisciplinary program and looked for those who best responded to these therapies [25]. Among the various results examined, it is interesting to note that the most frequent cause of treatment suspension was not the presence of side effects, but simple advice from another person that induced a change/the termination of prophylaxis. Furthermore, Harpole and colleagues performed a prospective analysis of the multidisciplinary treatment program for patients with migraine, Tension Type Headache (TTH), MOH, cluster headache and other headache entities, involving headache specialists, psychologists and primary care physicians. They observed a reduction of 21.2 points in Migraine Disability Assessment (MIDAS) after 6 months, including a reduction of 14.5 headache days on average within 3 months. The percentage of patients with a reduction of at least 50% in headache frequency was not calculated. In these conditions, a nursing intervention could have a significant effect on maintaining adherence to the therapeutic program [24].

A reduction in the relapse rate and greater effectiveness in suspending the continuous use of anti-migraine drugs was observed in a study performed in a hospital setting when associated with a psychological support program intervention [53]. In a follow-up study, a multidisciplinary approach proved effective for patients resistant to de-addiction therapies, and, after one year of combined analgesic therapy following withdrawal from antimigraine medications, over 80% of patients lost the diagnosis of MOH and half of the cases reported a return to forms of headache but with a reduction of more than 50% in the number of headache days experienced. The preparation and motivation of operators is essential to improve responses in cases of medication overuse in adolescents [54].

Cognitive behavioral therapy has also been shown to be useful in the discontinuation of excessive drug use, as well as other complementary actions such as relaxation exercises, autogenic training, controlled physical activity, lifestyle modifications, biofeedback and stress management, which may also be useful in the management of MOH [39].

The introduction of a specific therapy managed by a trained nurse produced the following results: the group supported by a headache nurse showed a significant improvement compared to non-supported patients regarding a reduction in overused drugs (73.1% vs. 60.7%; *p* = 0.008, respectively), and this result was confirmed by a multivariate analysis (Odds Ratio 1.73, 95% CI 1.11–2.71, *p* = 0.016). On the other hand, support from a headache nurse did not directly affect the clinical response, whereas this response was correlated with the underlying primary headache [55]. Behavioral treatments in the management of primary headaches in adults and children are increasingly recognized as effective; however, the level and duration of their effectiveness is still a matter of debate. In particular, the study delved into emerging behavioral treatments: biofeedback, music therapy and behavioral therapy, which produce considerable effects [56].

In a retrospective study of over 400 patients with MOH in Leiden, Pijpers and colleagues followed a study protocol that entailed the simple suspension of any treatment, in addition to the overused drug, and enrollment in the study protocol was carried out only for patients who had completed all three months of the suspension. Patients received advice on withdrawing from all acute headache medication (triptans, analgesics, a combination of both, other medications comprising opioids and ergots or combinations of those medications with analgesics or triptans), prophylactic medication and caffeine-containing liquids for two or three months. Follow-up occurred after withdrawal, to determine the final underlying primary headache diagnosis and start further treatment [57].

No significant differences were observed when comparing the effectiveness of counseling interventions for the discontinuation of excessive medication use, both in out- and in-hospital patients [58].

Cognitive behavioral therapy plus pharmacotherapy was not superior to pharmacotherapy alone in the discontinuation process at 6 and 12 month follow-ups, although psychotherapy is considered to be an effective method when added to pharmacotherapy for MOH, in order to prevent relapse [59].

Taking all these studies into consideration, we can underline the use of a variety of methodologies not directly comparable with each other that produced conflicting results; therefore, overall these studies suggest that counseling interventions demonstrate a rather modest effectiveness in resolving headache and preventing relapse in patients with MOH.

### 3.7. Onabotulinum Toxin A

The original studies on the use of onabotulinum toxin A (BT-A) in the prevention of chronic migration were introduced a decade ago, with the protocols PREEMPT 1 and 2, which enrolled two-thirds of patients with MOH and showed a significant reduction in the number of headache days and drug intake [60]. It is interesting to note that these patients did not follow the protocol of withdrawal from overused drugs, and the improvement was obtained starting from the first three months of therapy [61].

In contrast, there was a study comparing the efficacy, at 12 weeks, of add-on BT-A in patients with MOH who had suspended their drug consumption before starting BT-A, which did not demonstrate an additional benefit when compared to the simple drug withdrawal [62]. Subsequently, the same authors carried out a new prospective study that compared a nurse-intensive contact schedule with a minimal contact schedule, and this treatment arm was concealed inside another trial investigating BT-A. The endpoints were successful withdrawal and days with ingestion of migraine drugs after the withdrawal period. At three months, 179 patients showed a modest reduction in painkiller use after an intensive schedule, but there was no reduction in the number of headache days, confirming that the major efficacy of nurse behavioral intervention was maintaining patients’ adherence to treatment [57].

On the other hand, this drug is one of the few drugs for which approval has been given for use as a prophylactic drug in chronic migraine, even with MOH. Real life studies have confirmed the effectiveness of BT-A in the treatment and long-term management of chronic migraine [8]. In recent times, numerous studies have been carried out which have made the mechanism of action of BT-A in the prevention of chronic pain clearer and more convincing, although conclusive studies on the exact mechanism of action in humans, and in particular in chronic pain, are not still available [63]. However, clinical studies have confirmed the effectiveness of BT-A in chronic migraine, confirming it to be a safe and better-tolerated drug than traditional preventive drugs. Even if its effectiveness in MOH has been reported in real-life studies, we must find definitive confirmation [64].

## 4. Discussion

### 4.1. Management

Since the end of the last century there has been much discussion about the role of nurses in headache centers, especially the functions of nurses who work in first-level and third-level headache centers. This debate has developed under the pressure of new specific therapies for the treatment of migraine and the role that nurses have assumed in patient management and in participation in clinical research, in step with the increasing academic role of the discipline. Furthermore, over time, the principle of valorizing the more global parameters in the evaluation of the patient’s well-being, in addition to the classic parameters of pain and analgesic consumption, has been established. The complexity of the management of therapies and clinical studies has therefore made it necessary for nurses to increasingly commit to optimizing the organization of the department and taking care of the collection of clinical data and the management of chronic patients.

### 4.2. Nurses and MOH

In this particular sector, patients with MOH represent an important specific sample, due to the relevance of the cases in third-level centers and, conversely, in the management of these chronic patients at home. In 2011, Gaul and colleagues reported that 12 headache nurses were on duty in neurology services and that numerous patients were given nurse-led care following their diagnosis, which was performed by doctors. These nurses had followed a formalized course and had completed an online nursing course in the UK delivered through the Migraine Trust. Many European countries are developing similar courses for headache specialist nurses, and it would be hoped that the International Forum of Headache Nurses could, in the near future, help develop educational services and programs throughout Europe [28].

The activities they can carry out also include consultations with patients to monitor their progress, monitoring the effectiveness of drugs and supporting patients in therapeutic changes, from the assessment to the evaluation of their disability and advice on lifestyle issues. Therefore, the role of the nurse is effective, since they can improve the overall experience of the patient and the organization [25].

### 4.3. Multidisciplinary

Specialist nursing activities seem to be able to really improve the organization of the departments and the experience of patients in headache clinics, as they are able to influence, overall, all the parameters related to the use of the service. Furthermore, nurses who participate in research and training will be encouraged to improve the services provided as a whole, with attention paid to patients’ needs, the introduction of new therapies and the curiosity to monitor the real effectiveness of these treatments [28].

A study by Gordon and colleagues suggested a multidisciplinary, interdisciplinary and transdisciplinary team approach to pain management at New York University’s Rusk Institute of Rehabilitation Medicine Langone Medical Center. This approach emphasized mutual learning, training and education and the flexible exchange of specific disciplinary roles. Clinicians could implement a unified, holistic and integrated treatment plan with all team members accountable for the same patient-centered goals. The model described in the article promotes and reinforces the patient and family goals of the support system within a socio-cultural context [65].

A prospective controlled study compared the impact of a combined medical and nursing approach, lasting one year, and examined the effect of the nurse on classic headache outcomes and quality of life. This study showed that a brief nursing intervention based on lifestyle adjustments, combined with telephone support, improved the functionality and self-efficacy of migraineurs [66].

### 4.4. Nurses and Chronicization

The topic was already discussed few years ago, when the need to introduce into the evaluation of headaches the emotional and behavioral aspects, which significantly influence the experience of pain, and also the ability to express the pathology itself, was recognized [67,68].

In headache centers, headache nurses can play a particular role in following-up with patients, mainly sufferers of chronic headaches and/or MOH, assessing the quality of life and the level of disability and can provide support and advice with regard to lifestyle and self-management of drugs. Moreover, there are some examples of a handful of entirely nurse-led clinics where nurses diagnose, prescribe medications and treatments such as BT-A and peripheral nerve blocks, educate patients and follow-up on their care, and the results show these clinics are safe and cost-effective [67,69]. Moreover, Katsuki and colleagues conducted a campaign, through two main interventions, to evaluate knowledge about headache awareness in a student population and achieved outstanding results by undertaking this during mass COVID vaccination. This represents an interesting result, although it is not repeatable given the conditions of achievement, but it highlights the fundamental role of nurses in achieving and increasing disease awareness in a public health setting [70].

This therapeutic approach directly calls into play the functions of nursing, both in the diagnosis and in the treatment and management of chronic patients, even if, to date, there are no studies aimed at exploring this aspect. In this direction, an e-Delphi study was performed to know the opinions of neurologists and nurses in Europe on this topic [38].

Similar conclusions have also been reached in a USA update work, published by the National Institute of Health. Since the diagnosis and management of patients with MOH is complex, it is advisable to organize specialist teams, including internal medicine doctors, neurologists, pharmacists and trained nurses, to pursue indispensable objectives such as the education of patients and families on the non-problematic use of painkillers. Possible underlying psychiatric pathologies must also be carefully assessed and treated appropriately. These multi-professional teams must operate jointly, each reporting the actions related to their profession, with the purpose of detecting changes in the patient and correcting them if necessary. This is necessary because about 50% of patients relapse into overuse within 5 years, so constant monitoring seems to be an essential part of the therapy, as “there are no specific guidelines, but some professionals suggest tapering prophylactic medication after one year” [19].

### 4.5. Limitations

The main limitations of this research are linked to the fact that, from the large number of articles published on the topic of MOH, only a few examine the role and functions of nurses in this sector, while always expressing an appreciation of their professionalism and hoping for an increase in the relevance of the professional role. A second limitation is linked to the fact that the definition of the profession differs in different countries, so the term specialist nurse is present in some countries and not in others; additionally, specialization in headaches does not exist in Europe. Nonetheless, the treatment of MOH requires specific skills that straddle pharmacology, toxicology, neurology and psychiatry. Table 1 summarizes the results of past studies evaluating the nursing role in different headache centers. The heterogeneous sample and the methodology adopted to treat MOH called into question the demonstration of a causal relationship between the drugs used and the worsening of headache. Future studies should be performed with a more rigorous research methodology and a structured design that can allow for the acquisition of evidence that has greater scientific validity and is therefore useful for the creation of generalizable guidelines.

## 5. Conclusions and Future Directions

From the overall analysis of the literature examined, some conclusions can be drawn and summarized for a rational approach to patients with complex pathologies, such as chronic headache and MOH.

Patient education is crucial for efficient management to reach a true patient-centric approach, as well as for introducing nurses to the compilation and management of treatment plans that precisely define the role of patients in improving their education in understanding problems related to their health.

Uncomplicated MOH can be managed in a primary care setting, whereas complicated cases should be managed by a multidisciplinary team, mainly in the hospital setting.

Patients with poor education or who are unable to manage headache and overuse need to be treated with a detoxification and prevention program tailored for their clinical and socio-economic conditions. For this purpose, the discontinuation of the drug in question should be carried out immediately for patients using simple analgesics, such as ergot preparations and triptans. In the case of continuous use of opioids, barbiturates and sedatives, gradual suspension is recommended, preferably in a protected setting. Suspension can be carried out in various clinical situations, but it is always necessary for there to be a multidisciplinary team that can support staff and patients in this delicate phase of treatment.

Ultimately, the evidence emerging from the literature suggests that a holistic and multi-professional teamwork approach must be used for patients with MOH because this pathology has close and direct relationships with their psychopathological and biosocial state, such as the eating behaviors of the patients.

In the end, it is important to keep in mind that the interventions’ effectiveness in the medium and long term are similar in all these conditions, and this reinforces the need to use individualized therapy [14].

From the current literature, we can highlight a number of articles that address directions for the development of nursing in headache centers and in the management of MOH in the future.

Last year, Rasmussen tried to verify the opinions of various professionals involved in migraine treatment in Europe. This e-Delphi study could pave the way for implementing nurses’ work plans and ways to facilitate the sharing of information with patients regarding their clinical fate, as well as assessments of nursing care activity in the clinic. In the future, these indications could form the basis of a shared European protocol on nursing tasks in the treatment of migraine, based on scientifically valid evidence. These recommendations could favor further research that will have positive impacts on the coordination of medical staff to improve nurse care methodology oriented towards the well-being of migraine patients and suggest ideas and organizational models to be submitted to opinion- and policy makers of European healthcare organizations [38].

In the same period, Leonardi, on behalf of a large staff of experts from the World Health Organization (WHO), developed an operational plan to intervene in specific areas, including headaches [71].

The WHO established the “World Health Organization Intersectoral Global Action Plan on Epilepsy and Other Neurological Disorders 2022–2031 (WHOiGAP)” and, in May 2022, the 75th World Health Assembly endorsed this plan (World Health Organization, Geneva, 2023). In this congress, the aim of the plan was to move from global failure in reducing the burden of headache to triggering some specific strikes, so the WHOiGAP stated the following objectives to be achieved by measurable methods and in defined time periods: (1) increase political priority and strengthen governance; (2) provide effective diagnosis, treatment and care in a timely manner; (3) implement promotion and prevention strategies; and (4) promote research and innovation and manage information strategies.

The possible roles of nursing are reported in strategic objective 2: improving outcomes for cephalagic patients will mainly depend on the better organization of primary care, whether provided by general practitioners, clinicians, pharmacists or nurses. However, it is important to underline the urgency of this intervention [71].

This WHOiGAP could truly be a revolutionary approach in addressing neurological problems, with the dual objective of exploiting the increased attention of policy makers on neurological problems and, at the same time, proposing tools for this purpose, both an increase in resources and in personnel that must be trained.

The authors concluded that: “The community of clinicians and researchers working in the field of headache disorders must be part of this revolution to transform headache from an invisible disorder to a visible one in the public health scenarios. The chance to act is here and now: achieving this aim means improving the health and well-being of people with headaches. The headache revolution has already begun and must go on” [71].

Regarding nursing sciences, this revolution appears necessary and urgent with a view to optimizing nursing resources and skills, especially in primary care systems, but also in specialist centers. In this way, we could begin to complete the revolution in public care systems, which began over a century ago and must find its way through the commitment and direct involvement of the European Nurse Council in the implementation of projects, such as those proposed by WHOiGAP.

As an example, in Italy, the healthcare system should focus and direct efforts in the primary care departments by using the case manager organizational model, which is widely used by nurses experienced in home care or mental health. Furthermore, through the family and community nurses, who are experts in assessment and care planning, critical situations that have previously been underestimated could be intercepted early. During care planning, the nurse can make use of the nursing diagnosis system shared with the specialist doctor and the general practitioner, in order to obtain optimal results for the psycho-physical and social well-being of the person suffering from headache and of their family, which are crucial for compliance and the management of therapy [72].

Finally, one fundamental aspect emerging repeatedly over the time span covered by this survey is that the therapeutic approach must be multidisciplinary, and this aspect requires a considerable organizational effort towards management and patient involvement, both in and out of the health services. The role of adequately trained nurses therefore appears absolutely preeminent and must be taken into consideration when planning the development of intervention strategies in this type of chronic disease, which induces such great suffering among the population.

## Figures and Tables

**Table 1 brainsci-14-00600-t001:** Some examples of personal organization in tertiary headache centers.

Headache Center	Ref.	Nurse Training	Nurse on Staff	Neurology	Physical Treatments	Psychology
Calgary	[21]	NA	+	+	+	+
Copenhagen	[22]	Repeated visits	+	+	+	+
Copenhagen	[23]	Group session + visits	+	+	+	+
Durham	[24]	Regular visits	+	+		
Essen	[25]	36 h	+	+	+	+
Copenhagen	[6]	NA	+	+	+	+
Emilia-Romagna (Italy)	[26]	NA	+	+	+	+
Pavia (Italy)	[27]	Training	+	+	+	+

Organization of a level 3 headache center: nursing role, and other organization characteristics. Modified from [28].

## Data Availability

Not applicable.

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
