# Peer review of "The Nursing Role in the Management of Medication Overuse Headache: Realities and Prospects"

_brainsci, 2024, doi:10.3390/brainsci14060600_

Round 1

Reviewer 1 Report

Comments and Suggestions for Authors

Thank you for your good report. This report will improve nurse's activity on MOH.

Abst

Check space is corrected. Too much space is observed.

1. How to use template

Delete

Intro

ICHD-3, not ICHD only.

Please describe the criteria of MOH in ICHD-3. What headache should be MOH and why nurse education is needed?

4.Guidlines and statement

The information is very fragmented and difficult to read. Please put it in a table or organize it better. Especially, what has been done in the past and what is being done now are mixed up when just reading this.

Discussion

Please add subheadings for clarity.

There have been successful large-scale awareness-raising campaigns on MOH. Three other such efforts are underway in other countries. Please also describe how nurses can be involved in raising awareness in their communities.PMID: 36705435

Author Response

Reply to the Reviewer1

First of all, we would like to thank the Reviewer for his kind comments and the hope that this work can improve the nurse’s activity on MOH.

1 Abstract                         We cheeked spaces and corrected it

2 Template                       This paragraph has been removed

3 Introduction                 ICHC-3 corrected and a definition of the MOH has been included

4 Guidelines and…       The title of the section has been changed and the text has been modified in accordance with the Reviewer's suggestions.

5 Discussion                    The campaign of awareness-raising on MOH has been introduced in this section and discussed.

Reviewer 2 Report

Comments and Suggestions for Authors

Medication overuse headache (MOH) is a major problem and the authors here have searched to identify articles related to headaches and nursing management, and how non-pharmacological approaches to pain management, such as multidisciplinary team approaches, holistic treatment, cognitive behavioral therapy, and exercise can be used. Pubmed, Cinahl, Scopus, Embase, in the period between 2000 and 2024 searched. Based on the available evidence, the authors have presented nursing assessment and assistance strategies to plan an individual path relating to the needs of patients.

The authors are encouraged to elaborate on the following points in the revised version:

1.       Please remove the first paragraph, which is a part of the template, and start from the introduction as the first section:

1. How to Use This Template

The template details the sections that can be used in a manuscript. Note that each section has a corresponding style, which can be found in the “Styles” menu of Word. Sections that are not mandatory are listed as such. The section titles given are for articles. Review papers and other article types have a more flexible structure.

Remove this paragraph and start section numbering with 1. For any questions, please contact the editorial office of the journal or [email protected].

2.       Please check if the reference style matches the MDPI style.

3.       Please add the limitations of this review and sources of bias that might influence the findings and interpretation.

4.       This review mainly focuses on the nurses' role. please emphasize that early in the abstract and introduction.

5.       The pharmacological section is very limited in the review, please elaborate. The role of nurses can be dosing and monitoring of intervals, and also management of side effects. Alternatively, if these are tasks of the pharmacists, must be added.

6.       The authors are encouraged to present a holistic approach to teamwork and collaborative health care in this regard, even though they focused on the role of nurses. This role cannot stand alone, in particular since the authors have considered a holistic approach and non-pharmacological treatments that need other specialties in the care path of these patients.

7.       It is important to add the patient-centric approach as well, and how can the treatment plan take into account the patient roles, and also education, or health literacy.

8.       The authors are expected to design an action plan and they have named some examples in Italy. can those suggestions be generalized? What are the hurdles for the implementation of some of the suggestions? please elaborate.  

Comments on the Quality of English Language

There are several places where the expression of English is not clear or nonacademic. Proofreading the text by a native speaker or asking for a language edit service will help for higher text quality and clarity. 

Author Response

Reply to the Reviewer 2

We thank the Reviewer for the timely suggestions which encourage us to elaborate the underlined points and will allow us to improve the revised version.

  • The paragraph has been removed
  • The references have been completely revised in accordance with the suggestions of the Journal to match the MDPI style.
  • A paragraph with “Limitations” has been added in the Discussion section.
  • We have added further underlining this concept in the abstract and introduction.
  • The pharmacological section has been deliberately limited because, despite the numerous works that study the effectiveness of different drugs in MOH, we have very little certain data. Furthermore, the role of nurses in pharmacological treatments is well structured and consolidated in clinical practice, but is diversified depending on the pharmacological characteristics of the product used. For this reason, it was not deemed necessary to expand this section. We have introduced a paragraph in section 3.3 “Nurses” where we recall the possibility of including the professional figure of the nurse right from the design phase of the research protocols for controlled clinical studies. Furthermore, we fully agree with the Reviewer that the tasks indicated are specific to nursing and not to the pharmacists.
  • Some sentences have been included that underline the holistic and multidisciplinary approach in these chronic patients.
  • This theme was taken up at various points in the review and a new sentence was inserted in the conclusion sections, to underline the ability of multidisciplinary teams to build therapeutic plans tailored to individual patients.
  • In paragraph 3.3 a sentence has been inserted which illustrates an example of the official inclusion of the nurse in the planning and management phase of Controlled Clinical Trials through the inclusion of this professional figure directly in the research protocol. The definition of development plans for the profession are suggested in the WHO project proposals and in the activities of the international bodies that manage the development of health policies, as reported in the Conclusion and Future Directions sections.
  • The text has been again edited by a native English speaker.

Round 2

Reviewer 1 Report

Comments and Suggestions for Authors

Thank you.